# Molecular Pathways Associated with Cold Tolerance in Grafted Cucumber (*Cucumis sativus* L.)

**DOI:** 10.3390/plants14243860

**Published:** 2025-12-18

**Authors:** Sudeep Pandey, Bijaya Sharma Subedi, Andrew B. Ogden

**Affiliations:** 1Department of Horticulture, University of Georgia, 1109 Experiment Street, Griffin, GA 30223, USA; sudeep.pandey@uga.edu (S.P.); bijaya.sharmasubedi@uga.edu (B.S.S.); 2Institute of Plant Breeding, Genetics, & Genomics, 1109 Experiment Street, Griffin, GA 30223, USA

**Keywords:** grafting, cold tolerance, rootstock, gene expression, transcriptional factors

## Abstract

Cold stress limits cucumber productivity, and grafting onto tolerant rootstocks offers a promising strategy for improving resilience. This study compared the responses of cucumber heterografts and self-grafts exposed to different cold temperatures, aiming to uncover the molecular basis of grafting-mediated tolerance. Morphological observations showed that grafting onto *Cucurbita ficifolia* and *C. maxima* X *C. moschata* cv. Tetsukabuto rootstocks improved plant growth under moderate cold, while extreme stress remained lethal. Transcriptome analysis revealed that heterografts displayed broader and more sustained differentially expressed genes than self-grafts. Gene ontology (GO) enrichment in heterografts indicated early activation of structural, regulatory, and metabolic processes, with continued enrichment at later stages. KEGG analysis highlighted plant hormone signaling as a central pathway modulated by heterografting, with selective regulation of auxin, ethylene, and ABA signaling. Heterografts activated key regulators, including *MAPK3*-like, *TIFY5A*, and *CPK28,* which were strongly expressed, alongside transcription factors from NAC, CAMTA, WRKY, and MYB families, suggesting coordinated regulation of cold-responsive networks. These results demonstrate that heterografting enhances cold tolerance by orchestrating multi-layered molecular responses, including hormone modulation, stress signaling, and transcriptional factors. This underscores the potential of grafting onto cold-tolerant rootstocks as a practical strategy for cucumber cultivation in cold-prone environments.

## 1. Introduction

Cucumber (*Cucumis sativus* L.) is an important vegetable crop in the United States, and its production in controlled environment agriculture (CEA) systems is increasing worldwide [1,2]. The optimum temperature required for cucumber production is 26–30 °C (80–85 °F), and cucumber plants do not tolerate cold temperatures [3,4]. Maintaining optimum temperatures in greenhouses during colder seasons increases energy costs for growers [5]. Grafting has been widely adopted in vegetable production under adverse environments that pose abiotic and biotic stresses to enhance disease resistance, stress tolerance, and overall plant vigor [6]. The ability to utilize root physiological tolerance to stress independently of scion characteristics has facilitated the application of grafting [7,8].

Grafting onto resistant rootstocks can help to combat soilborne diseases, promote drought tolerance, and enhance vigor and nutrient uptake in different vegetable crops [9,10,11,12,13]. Grafting allows translocation of RNA [14,15,16], proteins [17,18,19], and other small molecules from the rootstock to the scion, directly affecting scion physiology [20,21]. This mostly leads to enhanced vegetative growth, fruit quality, and higher yields [7]. While grafting has been extensively studied in cucurbits for fruit quality and resistance to abiotic stresses such as salinity and drought, relatively limited information is available on its role in enhancing cold tolerance in cucumber [8,10,13,22,23,24]. Recent studies have begun to explore the physiological and biochemical responses of grafted cucumbers under low-temperature stress, showing improvements in photosynthetic efficiency, antioxidant enzyme activity, and root vigor [25,26,27]. However, the molecular mechanisms underlying these improvements remain largely unexplored. There is a significant knowledge gap regarding gene expression changes and signaling pathways activated in grafted cucumber plants exposed to cold stress. Specifically, the roles of hormonal signaling, stress-responsive transcription factors, and the translocation of cold-responsive molecules from rootstock to scion are not well understood. Elucidating these molecular pathways is critical to leveraging grafting as a strategy for improving cold tolerance in cucumber, especially in energy-intensive CEA systems.

Transcription factors (TFs) are responsible for the regulation of specific genes and plant-specific signals associated with biochemical and biological processes [28]. TFs such as MYB, AP2/ERF, and WRKY families play pivotal roles in regulating plant responses to cold stress by modulating the expression of stress-responsive genes [29,30,31]. In cucumber, previous researchers demonstrated that cold treatment induced the upregulation of key TFs, including MYB44, ERF054, WRKY48, and WRKY70, all of which have been associated with enhanced abiotic stress tolerance in other crops [27,32]. Notably, grafting onto different rootstocks resulted in distinct TF expression patterns, such as the specific induction of WRKY70 in ‘Yunnan figleaf gourd’-grafted plants, highlighting that rootstock genotype can differentially influence the transcriptional landscape and, consequently, cold tolerance [27]. This indicates that grafting not only serves as a physical modification for stress adaptation but also acts at the molecular level to alter TF-mediated regulatory networks, ultimately enhancing cold stress resilience in plants [33].

The parthenocarpic cucumber ‘Diva’ is a popular greenhouse cucumber variety and is prone to cold stress, like most other cucumber cultivars. Grafting onto cold-tolerant cucurbit rootstocks has shown potential to enhance cold resistance in cucumbers, which could help growers reduce heating costs [25]. One of the rootstocks commonly used in cucumber is figleaf gourd (*Cucurbita ficifolia* Bouché), also known as Malabar gourd, chilacayote, or shark fin melon, which is a cool-season squash native to the high-altitude regions of the Andes Mountains that grows well at relatively low soil temperatures (minimum 15 °C) [25,34]. Tetsukabuto squash (*C. maxima × C. moschata*), also known as Japanese winter squash, a winter squash hybrid, is an excellent rootstock for grafting watermelon, melon, and cucumber, offering disease resistance (Cañizares and Goto, 1998; Mohamed et al., 2014) [35,36].

This study was conducted using both heterografts (cucumber grafted onto *C. ficifolia* and Tetsukabuto) and self-grafts (cucumber grafted onto itself) to assess the impact of grafting on cold stress tolerance. The primary objective of this study was to identify key genes and molecular pathways regulated in grafted plants exposed to low temperatures. By comparing heterografts and self-grafts, this study aimed to uncover the physiological advantages conferred by cold-tolerant rootstocks and elucidate the genetic mechanisms underlying improved stress resilience, ultimately providing insights into enhancing cucumber production in cool environments.

## 2. Materials and Methods

### 2.1. Plant Materials and Grafting

The commercial parthenocarpic cucumber variety, Diva, was used as the scion cultivar. Figleaf gourd and Tetsukabuto squash were used as the rootstocks, whereas Diva rootstocks were used for self-grafted control. Diva and Tetsukabuto seeds were obtained from Johnny’s Selected Seeds (Winslow, ME, USA), and figleaf gourd seeds were sourced from the True Leaf Market Seed Company (Salt Lake City, UT, USA). These seeds were all seeded in 50-cell plug trays in a greenhouse at the University of Georgia, Griffin, Georgia. The rootstocks were seeded 3 days before the scions were seeded. After germination, seedlings were fertilized by hand-watering fertigated water at a constant feed rate of 100 ppm-N with 17-4-17 fertilizer (N-P_2_O_5_-K_2_0).

The one-cotyledon grafting method was used with grafting clips to attach 15-day-old scions to 18-day-old rootstocks. Rootstock and scion seedlings with similar stem diameters were selected. An angled cut (45–60°) was made just above one cotyledon leaf of the rootstock, ensuring that the rootstock apical meristem was completely removed to prevent unwanted regrowth. Rootstock roots were incised. Scions were cut at matching angles below the cotyledons, and their root system was discarded. The scion and rootstock were carefully aligned and secured using a grafting clip and inserted into the growing medium in 50-cell plug trays containing Pro-Mix BX potting mix. Treatments consisted of *C. ficifolia* + Diva (FG), Tetsukabuto + Diva (TG), Diva + Diva (SG), along with an ungrafted (UG) control. After grafting, plants were placed in a growth chamber (healing chamber at 29 °C) at the Department of Entomology, University of Georgia, Griffin, Georgia. Grafted plants placed in a healing chamber were covered with humidity domes to create an environment with 100% relative humidity. Plug trays were placed into a solid-bottom tray containing a small amount of water to promote humidity and reduce drought stress. Moisture was maintained inside the domes by misting plants twice per day.

Complete darkness was maintained for the first 48 h to minimize water loss and promote graft union formation. From days 3 to 7, plants were gradually introduced to light (18 h light/6 h dark, ~100 µmol m^−2^ s^−1^) and humidity by venting the dome. Once healed, grafted plants were first acclimated to greenhouse conditions for 2 days and then transplanted with the graft union positioned above the soil line to prevent scion rooting. All plants were transplanted into 25 cm diameter plastic pots (depth of 23 cm) with Pro-mix BX general purpose growing media (Premier Tech Growers and Consumers, Quakertown, PA, USA) and exposed to different temperatures.

### 2.2. Experimental Design

The experiment was conducted using a split-plot design with three growth chambers, set for three different day/night temperatures. G1 (12/7 °C), G2 (18/12 °C), and G3 (24/18 °C) were the main plots, and grafting treatments were the subplots. Each growth chamber had three blocks, and each block consisted of 6 plants for each treatment (*n* = 18 plants per treatment per growth chamber). The fresh weight and dry weight of plant tissue above the soil were measured on day 21 (Appendix A).

### 2.3. Sample Collection and RNA Extraction

Since almost all the plants died in G1, leaf samples from only G2 and G3 were collected for RNA extraction (Appendix A). Leaf samples from each treatment were collected on day 0 (D0) and day 21 (D21) for total RNA extraction. Three biological replicates per treatment were used for sequencing from G2 and G3 at D0 and D21. Each biological replicate consisted of leaves from two plants. A total of 48 samples were collected for library preparation (4 treatments * 3 replications * 2 growth chambers * 2 time periods). The uppermost young leaves from each treatment plant were collected for RNA extraction. Total RNA of plants was extracted using the Spectrum^TM^ Plant Total RNA kit (Sigma-Aldrich, St. Louis, MO, USA) following the manufacturer’s instructions. Approximately 40 µL of RNA was shipped to Novogene Corporation Inc. (Sacramento, CA, USA) for library preparation, and the remaining total RNA was stored at −80 °C for validation.

### 2.4. cDNA Library Preparation and RNA Sequencing

cDNA synthesis, library construction (mRNA libraries), and sequencing were conducted by Novogene Corporation Inc. (Sacramento, CA, USA). Prior to library preparation, RNA quality was evaluated using the Agilent 2100 Bioanalyzer (Agilent Technologies, Santa Clara, CA, USA), and RNA integrity numbers (RIN) were determined for quality control. Libraries were prepared using the Illumina TruSeq RNA Sample Preparation Kit following the manufacturer’s protocol. In brief, poly(A)-enriched mRNA was fragmented and reverse-transcribed into first-strand cDNA using random primers and reverse transcriptase. Second-strand synthesis was carried out with DNA Polymerase I and RNase H. Following ligation of Illumina TruSeq LT adapters, the cDNA fragments were PCR-amplified with standard Illumina primers using a minimal number of cycles to generate the final libraries. Sequencing was performed on the Illumina NovaSeq 6000 platform with a paired-end 150 bp (PE150) strategy.

### 2.5. Transcriptome Analysis

Trimmed reads of samples were mapped to the reference cucumber transcriptome using Bowtie2 v2.5.2 with default mapping parameters [37]. Gene count estimates were derived from the mapped reads using RNA-Seq by Expectation Maximization (RSEM) v1.3.3. A custom R script was used to determine fragments per kilobase million (FPKM) across all samples on R v4.4.1 using the following R libraries: dplyr, tidyverse, and stringr. DESeq2 was used to measure differentially expressed genes (DEGs) by comparing the gene counts between FG and UG, TG and UG, and SG and UG, where genes that had a |log_2_ fold change (LFC)| ≥ 1 and a false discovery rate (FDR) ≤ 0.05 were classified as DEGs [38].

### 2.6. Functional Annotation

The DEGs were annotated and assigned to gene ontology (GO) classes using the annotations of cucumber genes from the published draft genome [39]. The GO terms were processed and visualized using the WEGO web tool [40]. GO terms down to levels 2 and 3 were analyzed. The DEGs were then mapped to the Kyoto Encyclopedia of Genes and Genomes (KEGG) pathway database (http://www.genome.jp/kegg/ (accessed on 21 February 2025)), and the genes were classified according to the pathway in which they are involved or the functions they perform. The *p*-value for significant enrichment in the KEGG pathways was calculated using the hypergeometric distribution test, and the *p*-value was then corrected using Benjamini and Hochberg’s multiple testing correction. The KEGG pathways were considered to be significantly enriched when the corrected *p*-value (FDR) was ≤0.05.

### 2.7. Transcription Factors

Transcription factor prediction and family analysis of genes were performed by using PlantTFDB 4.0 (http://planttfdb.cbi.pku.edu.cn/ (accessed on 25 July 2025)) for the unique genes identified in G3 at D21 in different treatments.

### 2.8. Validation of RNA Sequencing Data

Reverse transcription–quantitative PCR (RT-qPCR) was performed to validate the transcriptomic results. Ten differentially expressed genes (DEGs) were randomly selected for validation, and their sequences were retrieved using seqtk. Total RNA samples stored at −80 °C were used for cDNA synthesis with the GoScript™ Reverse Transcription System (Promega Corporation, Madison, WI, USA) following the manufacturer’s instructions. The resulting cDNA was diluted 20-fold prior to qPCR. Primers for the target DEGs were designed using the NCBI Primer-BLAST tool (https://www.ncbi.nlm.nih.gov/tools/primer-blast, accessed on 23 March 2025), and their sequences are provided in Appendix A. qPCR reactions were prepared in a total volume of 10 µL containing 5 µL of 2X GoTaq^®^ qPCR Master Mix (Promega, Madison, WI, USA), 0.3 µM of each primer, 1 µL of cDNA, and nuclease-free water. Amplification was carried out on a QuantStudio 3 System (Applied Biosystems, Thermo Fisher Scientific, Waltham, MA, USA) under the following cycling conditions: 95 °C for 3 min, followed by 40 cycles of 95 °C for 15 s and 60 °C for 1 min. A melt curve analysis was performed to confirm the specificity of amplification. Each reaction included three technical replicates per sample for both target and reference genes, along with a no-template control. Relative gene expression levels were calculated using the 2−ΔΔCt method, with normalization to the internal reference gene tubulin (*Tub*) [41].

## 3. Results

### 3.1. Summary of RNA Sequencing

Three biological replicates per treatment in each growth chamber, G2 and G3 at D0 and D21, were included, resulting in a total of 48 libraries being constructed. Total raw reads and trimmed reads generated from FG, TG, SG, and UG plants ranged from around 37 to 50 million pair reads. The percentage of trimmed reads mapped to the reference cucumber transcriptome ranged from 75% to 98%.

### 3.2. Overview of DEGs

The number of DEGs varied across different treatments (Appendix A). In G2, the total number of DEGs identified at D0 in FG, TG, and SG was 310, 230, and 487, respectively. At D21, the total number of DEGs identified in FG, TG, and SG was 389, 244, and 607, respectively (Figure 1a). In G3, the total number of DEGs identified at D0 in FG, TG, and SG was 1598, 529, and 258, respectively. At D21, the total number of DEGs identified in FG, TG, and SG was 510, 353, and 186, respectively (Figure 1b).

The number of DEGs was mostly underexpressed in G2 at D0 and D21 in all three treatments (Figure 1a). The number of underexpressed DEGs was higher than overexpressed DEGs in G3 at D0 in all three treatments. However, the number of overexpressed DEGs surpassed the number of underexpressed DEGs at D21 in FG and TG (Figure 1b).

### 3.3. Functional Annotation and GO Enrichment Analysis

Gene ontology provided context for the functionality of genes and comprised three level 1 categories: biological process, cellular component, and molecular function. GO enrichment analysis revealed distinct patterns of DEGs across rootstock types, growth chamber, and time points. In G2 at D0, GO enrichment analysis revealed significant enrichment of level 2 GO terms related to cellular processes in FG plants. At level 3, GO terms associated with the intracellular part and hydrolase activity were significantly enriched. However, no significantly enriched GO terms were identified at D21 (Figure 2a). In G3 at D0, several level 2 GO terms, including cell, cell part, organelle, organelle part, binding, cellular process, and positive regulation of biological process, were significantly enriched. At D21, level 2 GO terms such as cellular process and transcription regulator activity remained significantly enriched in FG plants. Furthermore, GO terms including DNA-binding transcription factor activity, organic substance metabolic process, and primary metabolic process were also significantly enriched at this time point (Figure 2b).

In G2 at D0, GO enrichment analysis revealed significant enrichment of level 2 GO terms related to cellular processes in TG plants. At level 3, GO terms associated with nitrogen compound metabolic processes were significantly enriched. At D21, level 2 GO terms such as cellular process, metabolic process, and transcription regulator activity remained significantly enriched. Furthermore, GO terms (level 3), including DNA-binding transcription factor activity, organic substance metabolic process, cellular metabolic process, and primary metabolic process, were also significantly enriched at this time point (Figure 3a). In G3 at D0, level 2 GO term protein-containing complex and level 3 GO term regulation of metabolic process were significantly enriched in TG plants. At D21, level 2 GO terms such as cellular process and catalytic activity remained significantly enriched. Furthermore, GO terms including endomembrane system, oxidoreductase activity, and cellular metabolic process were also significantly enriched at this time point (Figure 3b).

In G2 at D0, GO enrichment analysis revealed significant enrichment of level 2 GO terms related to cell, cell part, organelle, organelle part, protein-containing complex, metabolic process, and transcription regulator activity in SG plants. In addition, multiple level 3 GO terms across all three main GO categories (biological process, molecular function, and cellular component) showed significant enrichment. At D21, level 2 GO terms such as cell, cell part, and protein-containing complex remained significantly enriched. Furthermore, the GO term (level 3) endomembrane system was also significantly enriched at this time point (Figure 4a). In G3 at D0, level 2 GO term binding, transcription regulator activity, and cellular process were significantly enriched in SG plants. Furthermore, GO terms (level 3) for heterocyclic compound binding, organic cyclic compound binding, DNA-binding transcription factor activity, and nitrogen compound metabolic process were also significantly enriched at this time point. However, no significantly enriched GO terms were identified at D21 in G3 (Figure 4b).

### 3.4. KEGG Enrichment Analysis

KEGG pathway enrichment analysis revealed distinct patterns of pathway activation across rootstock types, temperature treatments, and time points. In FG plants, significant enrichment of multiple pathways was observed at D0 in G2, notably including plant hormone signal transduction and thermogenesis. No significantly enriched pathways were detected in this group at D21. In G3, a broader set of pathways was significantly enriched in FG plants at D0, including plant hormone signal transduction, thermogenesis, plant-pathogen interaction, biosynthesis of amino acids, and the mitogen-activated protein kinase (MAPK) signaling pathway. By D21, enrichment persisted in several key pathways, including plant hormone signal transduction, plant–pathogen interaction, and the MAPK signaling pathway, indicating sustained activation of stress and signaling responses in FG plants (Figure 5a). In contrast, TG plants exhibited a much lower number of significantly enriched pathways across both chambers and time points, highlighting a comparatively muted transcriptomic response. However, the plant hormone signal transduction pathway was consistently enriched at both temperatures for D0 and D21, suggesting its central role in early and late graft responses regardless of rootstock type (Figure 5b). In SG plants, a higher number of enriched pathways were identified at D0 in G2, while no significant pathway enrichment was observed at D21 in G3 (Figure 5c).

### 3.5. Genes Associated with Plant Hormone Signal Transduction Pathway

KEGG enrichment analysis consistently identified the plant hormone signal transduction pathway as one of the most significantly enriched pathways across graft combinations, time points, and temperature treatments. In FG plants in G2 at D0, hormone signaling genes were predominantly underexpressed, indicating suppression of auxin, ethylene, and abscisic acid (ABA) pathways. Notably, transcription factor *PIF3*, a central integrator of multiple hormone signals, was highly underexpressed (log_2_FC = −23.61), along with glutamine synthetase (log_2_FC = −23.66), suggesting a strong shift away from nitrogen metabolism and growth signaling at this stage. In G3 at D0, a similar pattern emerged with marked underexpression of *ARR11*, *AUX28*, *SAUR23*, and *SAPK2*, implicating reduced auxin and cytokinin sensitivity. In contrast, a few hormone-related genes, such as *GH3.1* (log_2_FC = 4.42) and phytosulfokine receptor 1-like (log_2_FC = 10.71), were overexpressed, suggesting potential compensatory or tissue-specific responses. By D21 in G3, there was a rebound in hormone signaling, with overexpression of *TIFY10a* (jasmonic acid, JA), *EIN3* (ethylene), *BES1* (brassinosteroid), and *TMK1* (auxin), indicating a restoration or activation of developmental hormone pathways (Table 1).

In TG plants in G2 at D0, DEGs reflected suppression of auxin and ABA signaling, with underexpression of *ARF18*, *SRK2I*, and *ABSCISIC ACID-INSENSITIVE 5-like protein 5*. *PIF3* and *TIFY6B* were also strongly underexpressed (log_2_FC ~ −13 to −23), as seen in the FG plants. Notably, *serine/threonine kinase BSK2* was overexpressed (log_2_FC = 7.55), suggesting some activation of brassinosteroid signaling under these conditions. In G2 at D21, *ARF19* and *ATPase 11* were overexpressed, whereas *auxin transporter-like protein 4* remained underexpressed, indicating partial activation of auxin signaling but possible impairment in transport. The overexpression of *tryptophan aminotransferase-related protein* (log_2_FC = 10.45) suggests a potential upstream auxin biosynthesis response. In G3 at D0, dramatic transcriptional changes were noted, including very strong overexpression of *ARF18* (log_2_FC = 22.70) and *SRK2H* (log_2_FC = 22.97), coupled with consistent underexpression of auxin-induced genes (*AUX28*), TIFYs, and glutamine synthetase. This points to a transcriptionally active hormone signaling environment, though the direction of expression suggests an early stress response. In G3 at D21, several hormone-related transcription factors were overexpressed, including *MYC2* (JA), *TIFY10a*, *EIN3*, and *IAA13*, along with the auxin biosynthesis gene *YUCCA4* (log_2_FC = 8.97). However, *ethylene receptor 2* and *auxin-induced protein 15A-like* were underexpressed, suggesting selective tuning of signaling components (Table 2).

SG plants in G2 at D0 showed similar suppression patterns to those observed in heterografts, with underexpression of *GH3.1*, *ERF C3*, *ABI5-like*, and *PIF3*. The expression of *glutamine synthetase* and *calmodulin* was also markedly reduced, consistent with stress-associated responses at D0. In G2 at D21, *ARF18*, *ARF19*, and *pathogenesis-related protein 1* were moderately overexpressed, suggesting partial activation of auxin and defense pathways. However, *EIN3-like* remained strongly underexpressed (log_2_FC = −10.31), indicating limited recovery of ethylene signaling. In G3 at D0, hormone signaling components such as *TIFY4B*, *ABI5-like*, *MYC3*, and *IAA29* were underexpressed, consistent with the early suppression observed in other graft combinations. However, *TIFY6B* was strongly overexpressed (log_2_FC = 10.61), which may reflect an early JA response unique to this treatment. At D21, no DEGs from the plant hormone signal transduction pathway met the threshold criteria in G3, suggesting either homeostasis was restored, or transcriptional activity had normalized by this time point in self-grafted plants (Table 3).

### 3.6. Common and Unique DEGs

To assess transcriptional changes induced by different rootstock types, DEGs for FG, TG, and SG plants in growth chambers G2 and G3 at D0 and D21 were compared. Regarding DEGs in G2 at D0, the number of unique DEGs was 150 for FG, 117 for TG, and 334 for SG plants. A total of 50 DEGs were shared among all three rootstock types, with additional pairwise overlaps observed between the treatments. At D21, the number of unique DEGs increased to 226 for FG, 171 for TG, and 516 for SG plants, with 19 DEGs commonly shared among all three treatments (Figure 6a). In G3, FG exhibited the highest number of unique DEGs at D0 (1431), followed by TG (372) and SG (174) plants. In contrast, the number of unique DEGs at D21 drastically reduced in FG plants, and the number of unique DEGs also reduced in TG and SG plants. A total of 15 DEGs were shared across all three rootstock types at D21, while pairwise overlaps between treatments at D21 reduced in comparison with D0 (Figure 6b).

### 3.7. DEGs Uniquely Associated with C. ficifolia and Tetsukabuto Rootstocks

Parsing out DEGs shared with SG plants enabled the identification of gene sets uniquely associated with each heterograft and rootstock at D0 and D21 in both growth chambers. These unique DEGs represent transcriptional responses specifically attributable to the graft combination rather than the grafting procedure itself. This unique gene set was further examined for functional enrichment and pathway associations.

Each grafting treatment displayed a distinct set of unique DEGs that reflected graft-specific early molecular responses in G2 at D0. More pathways were significantly enriched in SG compared with heterograft plants. However, most of the unique DEGs in these pathways in SG plants were underexpressed. In FG plants, unique DEGs were dominated by hormone signaling components, including strong underexpression of jasmonic acid–amino synthetase and L-tryptophan–pyruvate aminotransferase. TG plants exhibited underexpression of auxin response factors, SRK2 kinases, and JAZ proteins. SG plants showed significant underexpression of endoplasmic reticulum (ER) protein processing genes, including *BiP*, *hsp70*, *hsp 90*, and *DnaJ* family members and phenylpropanoid enzymes, while uniquely overexpressing photosynthetic light-harvesting and photosystem I components (Appendix A).

At D21 in G2, heterografts continued to show relatively few significantly enriched pathways compared with self-grafts. In FG plants, no pathways were significantly enriched. In TG plants, unique DEGs were predominantly associated with hormone signaling, including overexpression of L-tryptophan–pyruvate aminotransferase, auxin response factor, and H^+^-transporting ATPase, suggesting enhanced auxin and amino acid–related signaling. SG plants displayed a broader set of unique DEGs affecting hormone signaling, ER protein processing, secondary metabolism, and cofactor biosynthesis. Notably, S-specific DEGs included underexpression of ethylene-insensitive protein 3, jasmonate ZIM domain proteins, ABA-responsive element binding factors, serine/threonine-protein kinase *CTR1*, and several enzymes in phenylpropanoid and cofactor biosynthetic pathways, while select ER chaperones (BiP, protein disulfide-isomerases, HSP20) and glycerolipid metabolism genes were upregulated. Additional unique DEGs in S involved suppression of enzymes in riboflavin and menaquinone-specific pathways, reflecting altered redox and metabolic regulation (Appendix A).

Unique DEGs were mainly associated with hormone signaling and plant–pathogen interaction in FG plants in G3 at D0. Notably, upregulation of xyloglucan:xyloglucosyl transferase *TCH4*, *MYC2*, calcium-dependent protein kinase, *GH3* auxin-responsive genes, and the phytosulfokine receptor indicated activation of auxin- and peptide-mediated signaling. Several jasmonate ZIM domain proteins, SAUR family members, BR-signaling kinase, and ABA-related transcription factors were downregulated, suggesting suppression of specific hormone response elements. Unique DEGs also affected phenylpropanoid biosynthesis, ER protein processing, cysteine and methionine metabolism, amino acid biosynthesis, MAPK signaling, RNA degradation, and cofactor biosynthesis, with selective upregulation of heat shock proteins (*hsp20*, *hsp70*, *hsp90*) and ER chaperones. TG plants showed stronger overexpression of hormone-related unique DEGs, including auxin response factors and serine/threonine-protein kinases, alongside underexpression of jasmonate ZIM domain proteins and glutamine synthetase. Other T-specific DEGs involved circadian rhythm, glucagon signaling, and mRNA surveillance pathways, reflecting early transcriptional adjustments. SG plants displayed limited enrichment, with downregulation of ABA-responsive factors; auxin-responsive IAA proteins; pathogenesis-related proteins; and key enzymes in alpha-linolenic acid metabolism, fatty acid degradation, and amino acid biosynthesis (Figure 7 and Appendix A).

KEGG pathways associated with plant hormone signal transduction were significantly enriched in FG and TG plants in G3 on D21 (Figure 8a,b). Along with that, the MAPK signaling pathway and cysteine and methionine metabolism KEGG pathways were also significantly enriched in FG plants in G3 at D21 (Figure 8a). None of the KEGG pathways in SG plants at G3 at D21 were significantly enriched.

KEGG pathway enrichment analysis revealed that unique DEGs in FG in G3 at D21 were predominantly associated with plant hormone signal transduction, MAPK signaling, and cysteine/methionine metabolism. The unique DEGs associated with plant hormone signal transduction were overexpressed in FG plants; however, in TG plants, the unique DEGs in this KEGG pathway were both overexpressed and underexpressed (Table 4). Unique DEGs associated with the MAPK signaling pathway and cysteine and methionine metabolism were also overexpressed in FG plants (Table 4). In plant hormone signal transduction, several key regulators were upregulated, including *mitogen-activated protein kinase 3-like* (log_2_FC = 4.60), *protein TIFY 5A* (log_2_FC = 3.94), *calcium-dependent protein kinase 28* (log_2_FC = 1.70), and *probable protein phosphatase 2C 6* (log_2_FC = 2.10). The MAPK signaling pathway shared overlapping elements with hormone signaling, such as *MAPK3-like*, *MAPKK5*, and *WRKY26*, indicating crosstalk between these pathways. Cysteine and methionine metabolism genes showed strong upregulation, including *serine acetyltransferase 5* (log_2_FC = 10.45), *glutamate–cysteine ligase* (log_2_FC = 11.26), and *cystathionine β-lyase* (log_2_FC = 10.74), suggesting enhanced sulfur amino acid biosynthesis in FG plants. In TG, hormone signaling changes were marked by strong induction of auxin biosynthesis and signaling genes (*IAA13*, log_2_FC = 10.13; *YUCCA4*, log_2_FC = 8.97) and strong repression of *ethylene receptor 2* (log_2_FC = −10.22), indicating a shift toward auxin-mediated responses and potential suppression of ethylene signaling.

### 3.8. Transcriptional Factors Associated with Cold Tolerance Uniquely in C. ficifolia and Tetsukabuto Rootstocks

Transcription factor (TF) analysis revealed distinct expression patterns among FG, TG, and SG plants (Table 5). In FG plants, 30 TFs were differentially expressed, spanning major families such as bHLH, bZIP, C2H2, CAMTA, ERF, G2-like, GATA, GRAS, HB, MIKC_MADS, MYB, NAC, and WRKY. Notably, strong overexpression was observed in NAC domain-containing protein 40 (log_2_FC = 20.67), CAMTA5 (log_2_FC = 10.93), and several MYB and WRKY members, suggesting activation of stress response and developmental regulation pathways. Both overexpressed and underexpressed TFs were detected, including G2-like PHR1-like 2 (log_2_FC = −11.43) and MYB-related telomere repeat-binding factor 1 (log_2_FC = −7.80). In TG plants, 9 TFs were significantly altered, with enrichment in bHLH, C2H2, HD-ZIP, NAC, B3, TALE, and YABBY families. The highest overexpression occurred in NAC domain-containing protein 50 (log_2_FC = 10.30) and B3 domain-containing transcription repressor VAL2 (log_2_FC = 10.11). Underexpressed TFs included BEL1-like homeodomain protein 7 (log_2_FC = −10.98) and YABBY 5 (log_2_FC = −3.35).

In SG plants, eight TFs were differentially expressed, representing B3, E2F/DP, ERF, NF-YA, TALE, Trihelix, and Whirly families. Most DEGs in SG associated with TFs were underexpressed, unlike heterografts. The most prominent overexpression was in Trihelix ASR3 (log_2_FC = 10.67) and E2FE (log_2_FC = 8.47), whereas BEL1-like homeodomain protein 7 (log_2_FC = −12.06) and NF-YA1 (log_2_FC = −11.82) were strongly underexpressed.

### 3.9. Validation

The RNA sequencing-based differential gene expression results were validated via RT-qPCR on three randomly selected DEGs for FG and TG plants in G2 and G3 growth chambers on day 21 (n = 12), using the tubulin gene from the cucumber plant as an internal standard. The results showed that the expression trends obtained from RNA sequencing and RT-qPCR were highly congruent. Pearson’s correlation coefficient for transcriptome and qPCR analysis was 0.877 (Figure 9).

## 4. Discussion

This study examined the transcriptomic responses of cucumber heterografts and self-grafts under cold stress to uncover the molecular mechanisms underlying enhanced tolerance conferred by cold-tolerant rootstocks. Previous studies have demonstrated the beneficial effects of grafting on fruit quality and abiotic stress tolerance; however, the molecular basis of cold tolerance in grafted cucumber remains largely unexplored. Our results confirm that grafting onto *C. ficifolia* and Tetsukabuto rootstocks improved growth under sub-optimal temperatures (G3), with moderate effects in G2, although plants could not survive extreme cold in G1 (12/7 °C). These findings indicate that the molecular differences observed under moderate cold stress can help identify key transcriptional and regulatory mechanisms supporting cold tolerance.

Heterografted plants exhibited a broader transcriptomic response compared with self-grafted ones. The distribution of DEGs across two temperature treatments revealed distinct transcriptional strategies in heterografted versus self-grafted plants. In heterografted plants, a greater number of DEGs were identified in milder G3 conditions than in the colder G2 environment, indicating that moderate cold elicited a more extensive transcriptional adjustment. In self-grafted plants, however, more DEGs were detected in G2 than in G3. The DEG profile in G2 was dominated by widespread underexpression, suggesting that colder temperatures imposed stronger physiological constraints that suppressed gene expression rather than activating coordinated defense pathways. These contrasting patterns indicate that heterografts retain greater transcriptional responsiveness under moderate cold, while self-grafts exhibit a more stress-driven, repressive transcriptional state under lower temperatures. Together, the results highlight a rootstock-specific influence on the capacity for transcriptional reprogramming during cold stress. GO enrichment analysis revealed that in FG plants, early activation (D0) of cellular structure, transcription regulation, and metabolic process terms reflects priming for structural integrity and metabolic readiness. By D21, enrichment persisted in cellular and metabolic processes, suggesting sustained transcriptional plasticity under prolonged cold stress. In contrast, self-grafted plants (SG) showed narrower GO enrichment, with D21 predominantly reflecting specific metabolic or catalytic functions. Differential DEGs across graft combinations likely reflect rootstock-mediated modulation of hormone transport and metabolic signaling. These findings indicate that the rootstock cultivar shapes the breadth and sustainability of transcriptional adaptation to cold, consistent with previous observations that grafting responses are largely rootstock-specific and influenced by environmental conditions [22,26].

KEGG enrichment analysis revealed that plant hormone signal transduction is the central pathway modulated by heterografting. In FG plants, early suppression of auxin, ethylene, and ABA-related genes likely represents an initial stress-mitigation strategy, with partial recovery of auxin and metabolic signaling by D21 [42,43,44,45,46]. At D0, in both G2 and G3, strong underexpression of *PIF3*, *ARR11*, *AUX28*, *SAUR23*, and *glutamine synthetase* reflects a broad downregulation of auxin, cytokinin, and nitrogen-associated signaling, establishing a low-energy physiological state that prioritizes stress preparedness over growth. This early hormonal suppression is partially counterbalanced by targeted activation of *GH3.1* and *PSKR1-like*, which modulate auxin homeostasis and peptide-hormone signaling, suggesting localized compensation to maintain essential cellular functions during cold exposure. By D21 in G3, the transcriptional profile shifts toward reactivation of developmental and stress-integrated pathways, marked by overexpression of *TIFY10a*, *EIN3*, *BES1*, and *TMK1*. These regulators bridge jasmonic acid, ethylene, brassinosteroid, and auxin signaling, collectively reinitiating growth, enhancing cell wall remodeling, and stabilizing auxin perception as plants move into an acclimated state. Together, these patterns support a system-level model in which FG plants initially suppress hormone flux to conserve energy and mitigate cold-induced damage, followed by a coordinated resurgence of growth- and defense-associated signaling once stress acclimation is established.

TG plants exhibited both over- and underexpression in hormone signaling genes, including strong induction of auxin biosynthesis genes (*IAA13*, *YUCCA4*) and suppression of ethylene receptor 2, suggesting selective modulation of hormone-mediated growth [47,48]. In TG plants, hormone-related DEGs revealed early suppression followed by selective pathway reactivation. At D0 in G2, strong underexpression of *ARF18*, *SRK2I*, *ABI5-like*, *PIF3*, and *TIFY6B* indicated broad repression of auxin, ABA, and JA-mediated growth signaling, consistent with an initial stress-mitigation strategy, whereas overexpression of *BSK2* suggested compensatory activation of brassinosteroid-mediated stabilization pathways. By D21 in G2, partial restoration of auxin signaling emerged through increased expression of *ARF19*, *ATPase11*, and tryptophan aminotransferase-related protein, although persistent underexpression of auxin transporter-like protein 4 implied restricted hormone transport despite renewed biosynthesis. Transcriptional shifts were more pronounced in G3 at D0, and overexpression of *ARF18* and *SRK2H* contrasted with continued suppression of *AUX28*, TIFY genes, and glutamine synthetase, reflecting a state in which upstream hormone regulators were activated but downstream growth programs remained repressed. By D21, signaling networks reorganized toward integrated growth–defense coordination, marked by strong induction of *MYC2*, *TIFY10a*, *EIN3*, *IAA13*, and the auxin biosynthesis gene *YUCCA4*, while selective suppression of ethylene receptor 2 and auxin-induced protein 15A-like indicated fine-tuned modulation of ethylene and auxin sensitivity. Collectively, these patterns show that TG plants transition from early hormonal suppression to a selectively reactivated, multi-hormone signaling environment, enabling a balanced recovery of growth and adaptive cold resilience.

Together, the FG and TG profiles show that heterografting activates a broader, more strategically tuned hormonal landscape than self-grafting, characterized by early suppression of auxin, ABA, JA, and ethylene growth signals followed by strong, selective reactivation of key regulators such as *MYC2*, *TIFY10a*, *EIN3*, *IAA13*, and *YUCCA4*. This phased transition from initial stress stability to coordinated growth and defense mechanisms reflects a more flexible and responsive transcriptional network in heterografts. Such finely regulated signaling reinstates developmental programs while maintaining stress readiness, ultimately conferring superior growth performance and enhanced cold resilience compared to self-grafted plants.

Notably, FG plants also displayed enrichment in MAPK signaling and cysteine/methionine metabolism, with upregulation of MAPK3-like, MAPKK5, WRKY26, serine acetyltransferase 5, glutamate–cysteine ligase, and cystathionine β-lyase, indicating coordinated crosstalk between hormone and stress signaling pathways as well as enhanced sulfur amino acid biosynthesis [43]. No significant KEGG pathway enrichment in SG plants signifies that self-grafting triggers weaker molecular responses under cold stress. MAPK3 in particular plays a prominent role in cold tolerance, functioning as a positive regulator through interactions with multiple cold-responsive transcription factors and its ability to modulate protective metabolic processes [49]. The pronounced activation of MAPK-related genes in heterografted plants highlights that MAPK signaling acts as a central integrator connecting hormonal cues, calcium signaling, and transcriptional reprogramming to support cold resilience. MAPK modules such as MEKK1–MKK2–MPK4/6 rapidly respond to cold-induced membrane rigidification and osmotic imbalances through phosphorylation cascades that amplify stress signals [50]. These pathways ultimately modulate *CBF* transcription factors and their downstream *COR* genes, enabling the plant to establish protective metabolic and physiological adjustments. While MPK3 and MPK6 often act as negative regulators to prevent excessive CBF activation and maintain growth–stress balance, MPK4 typically functions as an enhancer of cold-responsive gene expression [49].

Enrichment of cysteine and methionine metabolism in FG plants further underscores the metabolic reinforcement induced by heterografting. Genes involved in sulfur amino acid biosynthesis, including serine acetyltransferase 5, glutamate–cysteine ligase, and cystathionine β-lyase, were markedly overexpressed. Increased synthesis of cysteine and methionine supports enhanced production of glutathione and other sulfur-containing metabolites that function in antioxidant defense, redox buffering, and protein stabilization under stress. This metabolic activation complements hormonal and MAPK signaling, supplying the biochemical resources necessary to sustain cellular homeostasis during cold stress [43].

Analysis of DEGs unique to heterografted plants at Day 21 in G3 revealed that heterografting orchestrates a targeted transcriptional reprogramming. FG plants demonstrated strong overexpression of regulators such as MAPK3-like, TIFY5A, and calcium-dependent protein kinase 28, integrating stress signaling and hormonal regulation. MAPK3 plays diverse roles in plant cold tolerance, acting as a positive regulator through its interactions with key factors and involvement in multiple signaling pathways that modulate cold-responsive genes and metabolic processes [49,51,52]. TIFY genes are reported to respond to cold stress via ABA-independent signaling pathways [53,54,55,56]. Calcium-dependent protein kinases (CPKs) constitute one of the earliest and most sensitive components of cold perception due to their ability to decode cold-triggered Ca^2+^ signatures [57]. Upon binding Ca^2+^, CPKs undergo conformational changes that activate kinase functions essential for propagation of cold signals. These kinases directly regulate ROS-generating enzymes, ion channels, and transcription factors, and they initiate crosstalk with MAPK cascades to coordinate downstream stress responses. In tomato and other model systems, CPKs activate MPK1/2 modules under cold conditions, illustrating a conserved upstream influence on MAPK pathways. During cold stress, plants trigger calcium (Ca^2+^) signals to activate Calcium-Dependent Protein Kinase 28 (CPK28), which phosphorylates NIN-LIKE PROTEIN 7 (NLP7), and switch on genes that enhance freezing tolerance [58,59]. The integration of these genes enhances stress signaling and hormonal regulation, ultimately benefiting cold tolerance. TG plants exhibited enhanced auxin-mediated responses and selective suppression of ethylene signaling, consistent with rootstock-specific adaptation strategies. These molecular signatures align with the observed physiological performance, where heterografted plants exhibited higher fresh and dry weight, suggesting that targeted modulation of hormone, signaling, and metabolic pathways underpins enhanced growth and cold resilience.

Heterografting also resulted in broader and more dynamic regulation of transcription factors (TFs) compared to self-grafting. FG plants differentially expressed 30 TFs across major families (NAC, WRKY, MYB, bHLH, CAMTA, MADS), including strong overexpression of NAC40 and CAMTA5, reflecting activation of stress-responsive and developmental pathways. TG plants showed selective overexpression of NAC50 and B3-VAL2, while SG plants exhibited only eight DEGs, mostly underexpressed, highlighting constrained transcriptional flexibility. Notably, several overexpressed TFs in heterografted plants are well-known regulators of stress and developmental responses. NAC proteins are key modulators of abiotic stress tolerance, controlling downstream genes involved in osmotic adjustment, reactive oxygen species scavenging, and hormone signaling [27,60,61]. CAMTA5 links calcium signaling to cold-responsive gene activation, and WRKY/MYB members contribute to balancing defense, metabolism, and growth under low temperatures [29,62]. FG plants displayed a far more robust and strategically coordinated transcription factor response compared to TG plants, highlighting their stronger regulatory advantage under cold stress. FG plants activated a wide array of 30 TFs across major stress-responsive families, with pronounced induction of *NAC40*, *CAMTA5*, MYB, and WRKY factors collectively amplifying Ca^2+^-dependent cold signaling, enhancing ROS detoxification capacity, and reinforcing developmental flexibility essential during chilling stress. This broad activation indicates a highly integrated network capable of rapidly restructuring stress, metabolic, and growth pathways. In contrast, TG plants exhibited a narrower response, with only nine TFs differentially expressed. Although *NAC50* and *VAL2* were strongly induced, the overall regulatory breadth was limited, and key developmental regulators such as *BEL1-like* and *YABBY5* were suppressed, suggesting a more constrained adaptive strategy. Thus, the extensive transcriptional reprogramming observed in FG plants, spanning multiple signaling hubs and stress-response modules, demonstrates a superior, more versatile regulatory architecture that likely contributes to their stronger cold resilience relative to TG plants. The coordinated expression of these TFs suggests that heterografts activate a complex regulatory network that primes the scion for improved cold tolerance, whereas self-grafted plants, with limited TF activation, may lack sufficient transcriptional flexibility to respond effectively to chilling stress.

Overall, the combined GO, KEGG, and TF analyses demonstrate that heterografting onto cold-tolerant rootstocks enhances cucumber resilience through multi-layered transcriptional reprogramming. Heterografts show early suppression and later recovery of key hormone and metabolic pathways, broad TF activation, and enhanced amino acid metabolism, providing a coordinated molecular basis for improved growth under sub-optimal temperatures. Self-grafted plants, while capable of some stress responses, display narrower and less coordinated transcriptional adaptation, highlighting the advantage of heterografting in cold-prone cultivation.

## 5. Conclusions

Heterografting onto cold-tolerant rootstocks reprograms early stress-sensing and signaling networks in cucumber, resulting in a broader and more coordinated transcriptional response under cold stress. Enhanced Ca^2+^ transients in heterografts are rapidly decoded by calcium-dependent kinases (e.g., CPK28) and CaM-regulated factors (e.g., CAMTA5), triggering early transcriptional activation of NLP7 and other cold-responsive regulators, while a MAPK3-centered cascade integrates these calcium signals with hormonal and stress pathways. Concurrent induction of TIFY/JAZ regulators modulates ABA-independent signaling, collectively forming a multilayered hub that primes the scion for maintaining metabolic activity and structural stability. This network is reinforced by strong activation of NAC, WRKY, MYB, and bHLH transcription factors, coordinating downstream responses in ROS detoxification, osmotic adjustment, and cell wall remodeling. Rootstock-mediated modulation of hormone pathways further fine-tunes these responses, with transient suppression of auxin, ethylene, and ABA signaling followed by selective recovery, including reactivation of auxin biosynthesis genes such as YUCCA4 and balanced expression of ethylene regulators. Enhanced cysteine/methionine and glutathione metabolism supports sustained growth and redox balance. Together, these interconnected signaling and transcriptional modules form a unified regulatory model in which cold-tolerant rootstocks orchestrate a resilient program that promotes improved growth and cold tolerance in grafted cucumber, with FG plants showing stronger and more flexible transcriptional adaptation than TG plants.

## Figures and Tables

**Figure 1 plants-14-03860-f001:**
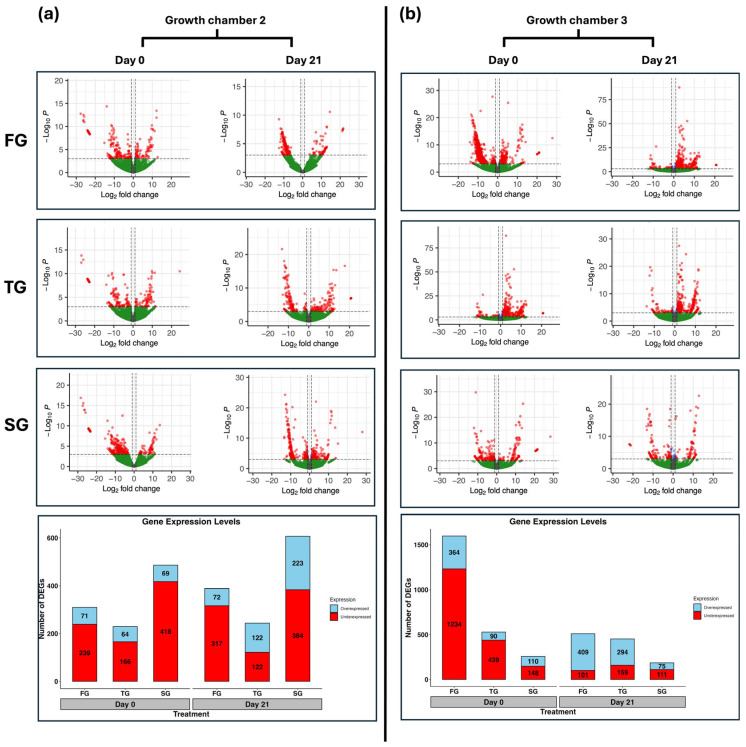
Volcano plots represent the differential expression profiles of hetero-grafted (FG and TG) and self-grafted (SG) cucumber plants on day 0 and day 21 in two growth chambers: (**a**) G2 (18/12 °C), and (**b**) G3 (24/18 °C). Stacked bar graphs represent the number of overexpressed and underexpressed DEGs in hetero-grafted and self-grafted cucumber plants on day 0 and day 21 in two growth chambers, (**a**) G2 and (**b**) G3. Genes with a log_2_ fold change |LFC| ≥ 1 and a false discovery rate (FDR) < 0.05 are highlighted in red and are differentially expressed. FG = cucumber plants grafted onto *C. ficifolia* rootstock; TG = cucumber plants grafted onto Tetsukabuto rootstock.

**Figure 2 plants-14-03860-f002:**
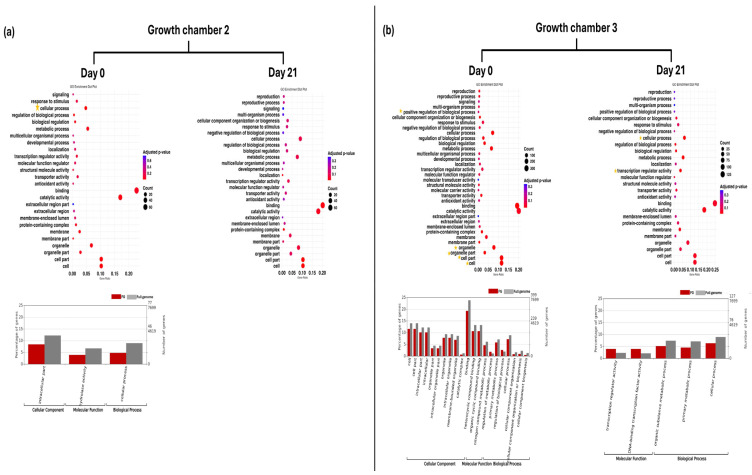
GO enrichment dot plot represents GO annotation analysis of DEGs for FG plants on day 0 and day 21 in two growth chambers, (**a**) G2 (18/12 °C) and (**b**) G3 (24/18 °C), at GO term level 2. Yellow stars indicated significantly enriched GO terms. Bar graphs represent GO annotation analysis of DEGs for FG plants on day 0 and day 21 in two growth chambers, (**a**) G2 and (**b**) G3, at GO term level 3. FG = cucumber plants grafted onto *C. ficifolia* rootstock.

**Figure 3 plants-14-03860-f003:**
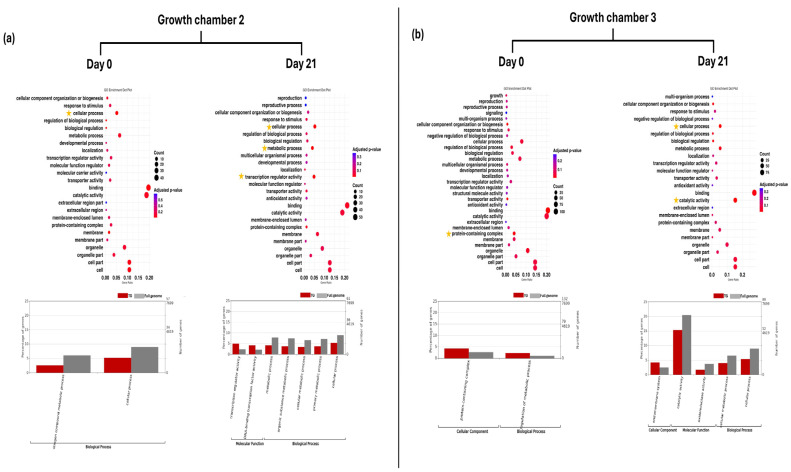
GO enrichment dot plot represents GO annotation analysis of DEGs for TG plants on day 0 and day 21 in two growth chambers, (**a**) G2 (18/12 °C) and (**b**) G3 (24/18 °C), at GO term level 2. Yellow stars indicated significantly enriched GO terms. Bar graphs represent GO annotation analysis of DEGs for FG plants on day 0 and day 21 in two growth chambers, (**a**) G2 and (**b**) G3, at GO term level 3. TG = cucumber plants grafted onto Tetsukabuto rootstock.

**Figure 4 plants-14-03860-f004:**
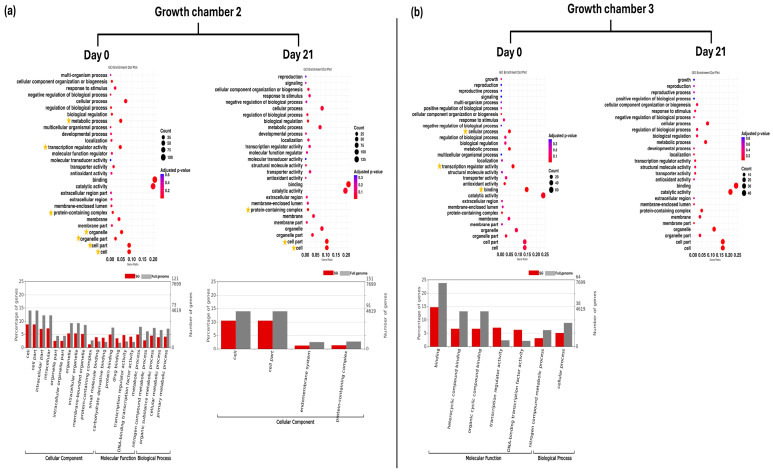
GO enrichment dot plot represents GO annotation analysis of DEGs for SH plants on day 0 and day 21 in two growth chambers, (**a**) G2 (18/12 °C) and (**b**) G3 (24/18 °C), at GO term level 2. Yellow stars indicated significantly enriched GO terms. Bar graphs represent GO annotation analysis of DEGs for FG plants on day 0 and day 21 in two growth chambers, (**a**) G2 and (**b**) G3, at GO term level 3. SG = self-grafted cucumber plants.

**Figure 5 plants-14-03860-f005:**
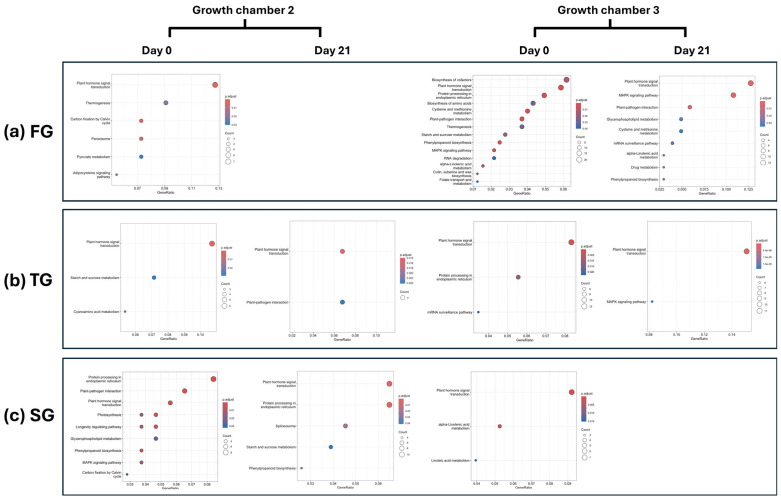
KEGG enrichment analysis of DEGs for (**a**) FG, (**b**) TG, and (**c**) SG plants on day 0 and day 21 in two growth chambers, G2 (18/12 °C), and G3 (24/18 °C). FG = cucumber plants grafted onto *C. ficifolia* rootstock, TG = cucumber plants grafted onto Tetsukabuto rootstock, and SG = self-grafted cucumber plants.

**Figure 6 plants-14-03860-f006:**
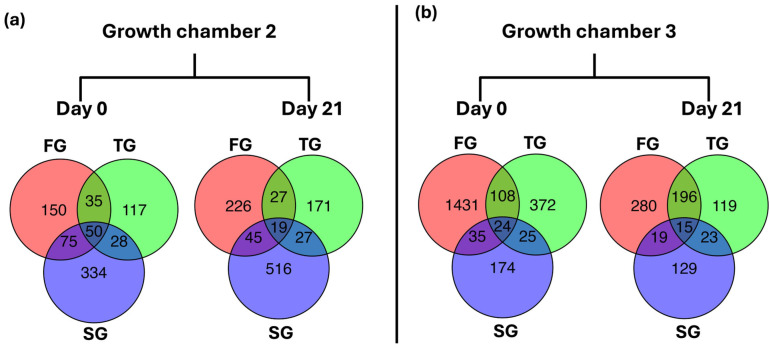
Venn diagram representing common and unique DEGS between FG, TG, and SG plants on day 0 and day 21 in two growth chambers, (**a**) G2 (18/12 °C) and (**b**) G3 (24/18 °C). FG = cucumber plants grafted onto *C. ficifolia* rootstock; TG = cucumber plants grafted onto Tetsukabuto rootstock; SG = self-grafted cucumber plants.

**Figure 7 plants-14-03860-f007:**
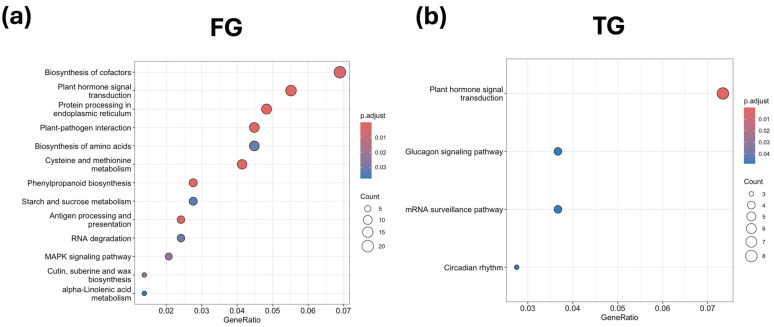
KEGG enrichment analysis of unique DEGs for (**a**) FG and (**b**) TG plants on day 0 in growth chamber G3 (24/18 °C). FG = cucumber plants grafted onto *C. ficifolia* rootstock; TG = cucumber plants grafted onto Tetsukabuto rootstock.

**Figure 8 plants-14-03860-f008:**
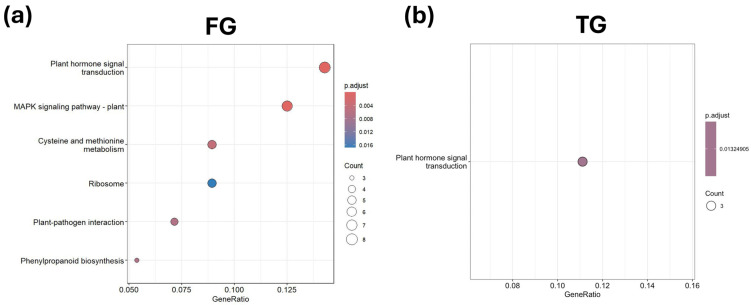
KEGG enrichment analysis of unique DEGs for (**a**) FG and (**b**) TG plants on day 21 in growth chamber G3 (24/18 °C). FG = cucumber plants grafted onto *C. ficifolia* rootstock; TG = cucumber plants grafted onto Tetsukabuto rootstock.

**Figure 9 plants-14-03860-f009:**
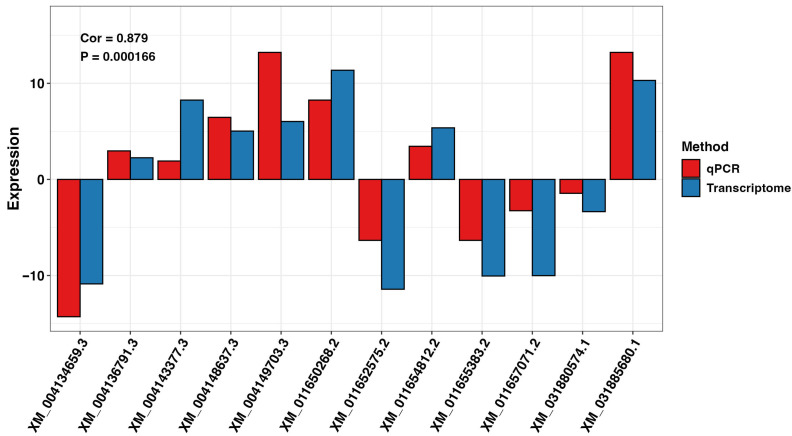
Comparison of expression levels of DEGs for FG and TG plants on day 0 and day 21 in growth chambers G2 (18/12 °C) and G3 (24/18 °C) analyzed using transcriptome and qPCR methods. FG = cucumber plants grafted onto *C. ficifolia* rootstock; TG = cucumber plants grafted onto Tetsukabuto rootstock.

**Table 1 plants-14-03860-t001:** Genes associated with plant hormone signal transduction pathway in FG plants.

	ID	Gene	Log_2_FC	Function
G2_D0	K14506	XP_004133822.1	−13.39	jasmonoyl--L-amino acid synthetase JAR6, transcript variant X1
	K14487	XP_004141893.1	−6.31	probable indole-3-acetic acid-amido synthetase GH3.1
	K14496	XP_004148737.1	2.19	abscisic acid receptor PYR1
	K14516	XP_004150983.1	−3.07	ethylene-response factor C3
	K01915	XP_011659421.1	−23.66	glutamine synthetase nodule isozyme, transcript variant X2
	K16903	XP_031736085.1	−23.31	tryptophan aminotransferase-related protein 2-like, transcript variant X3
	K12126	XP_031736743.1	−23.62	transcription factor PIF3, transcript variant X7
G3_D0	K14504	NP_001267579.1	3.22	probable xyloglucan endotransglucosylase/hydrolase protein 23-like
	K13422	XP_004133809.2	2.34	transcription factor MTB1
	K01915	XP_004134161.1	−10.38	glutamine synthetase leaf isozyme, chloroplastic, transcript variant X1
	K13464	XP_004137510.1	−11.53	protein TIFY 6B, transcript variant X1
	K13412	XP_004140192.1	3.31	calcium-dependent protein kinase 34
	K14505	XP_004140711.1	−10.64	cyclin-D3-3
	K14487	XP_004141994.1	4.42	probable indole-3-acetic acid-amido synthetase GH3.1
	K14491	XP_004142954.1	−8.09	two-component response regulator ARR11
	K14498	XP_004143455.1	−9.92	serine/threonine-protein kinase SAPK2, transcript variant X1
	K14484	XP_004145418.1	−9.10	auxin-induced protein AUX28
	K14488	XP_004147006.1	−8.81	auxin-responsive protein SAUR23
	K14500	XP_004147901.1	−9.30	serine/threonine-protein kinase BSK6
	K14497	XP_004148120.2	−1.58	probable protein phosphatase 2C 75, transcript variant X1
	K14492	XP_004149797.1	−6.65	two-component response regulator ORR9
	K12126	XP_011648885.1	−7.41	transcription factor PIF3, transcript variant X3
	K14431	XP_011655928.2	−2.36	transcription factor TGA2.2, transcript variant X7
	K13946	XP_011658595.1	−9.61	auxin transporter-like protein 5
	K27625	XP_031740874.1	10.72	phytosulfokine receptor 1-like
	K14489	XP_031742807.1	−10.73	histidine kinase 3, transcript variant X2
G3_D21	K14504	NP_001267579.1	3.92	probable xyloglucan endotransglucosylase/hydrolase protein 23-like
	K20536	NP_001267653.1	4.60	mitogen-activated protein kinase 3-like
	K13422	XP_004133809.2	1.56	transcription factor MTB1
	K13412	XP_004134863.1	1.70	calcium-dependent protein kinase 28
	K14497	XP_004135669.1	2.10	probable protein phosphatase 2C 6
	K14515	XP_004138725.1	1.53	EIN3-binding F-box protein 1
	K27628	XP_004140059.2	2.32	U-box domain-containing protein 15
	K00924	XP_004142822.1	1.04	receptor protein kinase TMK1
	K14503	XP_004143497.1	1.36	BES1/BZR1 homolog protein 2, transcript variant X1
	K14514	XP_004144109.2	0.68	protein ETHYLENE INSENSITIVE 3, transcript variant X1
	K13464	XP_004144433.1	3.93	protein TIFY 10a
	K13413	XP_004148562.2	1.07	mitogen-activated protein kinase kinase 5
	K14488	XP_031744790.1	−3.76	auxin-induced protein 15A-like

**Table 2 plants-14-03860-t002:** Genes associated with plant hormone signal transduction pathway in TG plants.

	ID	Gene	Log_2_FC	Function
G2_D0	K14500	XP_004133947.1	7.55	serine/threonine-protein kinase BSK2, transcript variant X2
	K14486	XP_004143252.1	−11.06	auxin response factor 18, transcript variant X1
	K14498	XP_004148038.1	−10.09	serine/threonine-protein kinase SRK2I, transcript variant X3
	K01915	XP_011659421.1	−23.31	glutamine synthetase nodule isozyme, transcript variant X2
	K12126	XP_031736743.1	−23.28	transcription factor PIF3, transcript variant X7
	K13464	XP_031741308.1	−13.91	protein TIFY 6B, transcript variant X4
G2_D21	K16903	XP_004138271.1	10.45	tryptophan aminotransferase-related protein 2-like, transcript variant X1
	K01535	XP_004148685.1	5.03	ATPase 11, plasma membrane-type
	K14486	XP_011648570.1	11.36	auxin response factor 19, transcript variant X1
	K13946	XP_011653685.1	−10.05	auxin transporter-like protein 4, transcript variant X1
G3_D0	K14486	NP_001295772.1	22.70	auxin response factor 18
	K14505	XP_004140711.1	−7.13	cyclin-D3-3
	K14484	XP_004145418.1	−8.42	auxin-induced protein AUX28
	K14431	XP_004149279.2	−7.59	transcription factor TGA2.3, transcript variant X1
	K14497	XP_004150316.2	−12.11	protein phosphatase 2C 51
	K14498	XP_011653912.1	22.97	serine/threonine-protein kinase SRK2H, transcript variant X1
	K13464	XP_011655950.1	−12.05	protein TIFY 6B, transcript variant X1
	K01915	XP_011657677.1	−9.36	type-1 glutamine synthetase 1, transcript variant X1
	K14489	XP_031742807.1	−10.13	histidine kinase 3, transcript variant X2
	K14500	XP_031743575.1	−10.22	serine/threonine-protein kinase BSK1, transcript variant X2
	K13412	XP_031745188.1	−8.80	calcium-dependent protein kinase 26, transcript variant X1
	K14490	XP_031745195.1	−8.53	histidine-containing phosphotransfer protein 4, transcript variant X2
G3_D21	K14504	NP_001267579.1	3.35	probable xyloglucan endotransglucosylase/hydrolase protein 23-like
	K14515	XP_004138725.1	1.26	EIN3-binding F-box protein 1
	K27628	XP_004140059.2	1.72	U-box domain-containing protein 15
	K14503	XP_004143497.1	1.00	BES1/BZR1 homolog protein 2, transcript variant X1
	K14514	XP_004144109.2	0.58	protein ETHYLENE INSENSITIVE 3, transcript variant X1
	K13464	XP_004144433.1	2.99	protein TIFY 10a
	K13422	XP_004148739.1	2.40	transcription factor MYC2
	K14484	XP_011649125.1	10.13	auxin-responsive protein IAA13, transcript variant X1
	K11816	XP_011649797.1	8.97	probable indole-3-pyruvate monooxygenase YUCCA4, transcript variant X1
	K14509	XP_031737453.1	−10.22	ethylene receptor 2, transcript variant X3
	K14488	XP_031744790.1	−5.04	auxin-induced protein 15A-like

**Table 3 plants-14-03860-t003:** Genes associated with plant hormone signal transduction pathway in SG plants.

	ID	Gene	Log_2_FC	Function
G2_D0	K14487	XP_004141893.1	−5.23	probable indole-3-acetic acid-amido synthetase GH3.1
	K14516	XP_004150983.1	−4.99	ethylene-response factor C3
	K14432	XP_011649580.1	−13.80	ABSCISIC ACID-INSENSITIVE 5-like protein 5, transcript variant X1
	K01915	XP_011659421.1	−23.96	glutamine synthetase nodule isozyme, transcript variant X2
	K12126	XP_031736743.1	−23.92	transcription factor PIF3, transcript variant X7
	K02183	XP_031739757.1	−13.47	calmodulin, transcript variant X3
G2_D21	K13449	XP_004136924.2	5.97	pathogenesis-related protein 1
	K14486	XP_004139643.1	2.33	auxin response factor 18, transcript variant X1
	K14514	XP_004140927.1	−10.31	ETHYLENE INSENSITIVE 3-like 1 protein, transcript variant X1
	K14488	XP_004147644.1	1.92	hypothetical protein
	K13464	XP_011649171.1	−9.53	protein TIFY 4B, transcript variant X3
	K14432	XP_011649580.1	−9.30	ABSCISIC ACID-INSENSITIVE 5-like protein 5, transcript variant X1
	K01915	XP_011651002.1	−1.57	glutamine synthetase leaf isozyme, chloroplastic, transcript variant X2
	K13946	XP_011653685.1	−10.20	auxin transporter-like protein 4, transcript variant X1
	K14431	XP_011655928.2	−2.43	transcription factor TGA2.2, transcript variant X7
	K14510	XP_011657691.1	−8.30	serine/threonine-protein kinase CTR1, transcript variant X3
G3_D0	K13449	XP_004136924.2	−4.04	pathogenesis-related protein 1
	K14432	XP_004144092.1	−8.88	ABSCISIC ACID-INSENSITIVE 5-like protein 2, transcript variant X4
	K13422	XP_004146202.3	−7.89	transcription factor MYC3, transcript variant X2
	K14431	XP_004149279.2	−8.05	transcription factor TGA2.3, transcript variant X1
	K14484	XP_004150206.2	−3.10	auxin-responsive protein IAA29
	K13464	XP_011650896.1	10.61	protein TIFY 6B, transcript variant X2
	K27627	XP_011651990.1	−2.20	phytosulfokines 5

**Table 4 plants-14-03860-t004:** KEGG enriched pathways associated with unique DEGs in FG and TG plants.

**FG**	**ID**	**Gene**	**Log_2_FC**	**Function**
Plant hormone signal transduction	K20536	NP_001267653.1	4.60	mitogen-activated protein kinase 3-like
	K14504	NP_001267702.1	6.33	probable xyloglucan endotransglucosylase/hydrolase protein 23-like
	K13422	XP_004133809.2	1.56	transcription factor MTB1
	K13412	XP_004134863.1	1.70	calcium-dependent protein kinase 28
	K14497	XP_004135669.1	2.10	probable protein phosphatase 2C 6
	K00924	XP_004142822.1	1.04	receptor protein kinase TMK1
	K13413	XP_004148562.2	1.07	mitogen-activated protein kinase kinase 5
	K13464	XP_004149668.1	3.94	protein TIFY 5A
MAPK signaling pathway	K20536	NP_001267653.1	4.60	mitogen-activated protein kinase 3-like
	K13424	NP_001292676.1	7.92	probable WRKY transcription factor 26
	K13422	XP_004133809.2	1.56	transcription factor MTB1
	K14497	XP_004135669.1	2.10	probable protein phosphatase 2C 6
	K20604	XP_004137516.1	2.32	mitogen-activated protein kinase kinase 9
	K13413	XP_004148562.2	1.07	mitogen-activated protein kinase kinase 5
	K20557	XP_011658569.1	1.61	uncharacterized LOC101216189
Cysteine and methionine metabolism	K08967	XP_004134118.1	0.79	1,2-dihydroxy-3-keto-5-methylthiopentene dioxygenase 2
	K27857	XP_004136690.1	1.13	cystathionine gamma-synthase 1, chloroplastic
	K00640	XP_011649387.1	10.45	serine acetyltransferase 5, transcript variant X2
	K01919	XP_031740812.1	11.26	glutamate--cysteine ligase, chloroplastic, transcript variant X2
	K01760	XP_031743082.1	10.74	cystathionine beta-lyase, chloroplastic, transcript variant X5
**TG**	**ID**	**Gene**	**Log_2_FC**	**Function**
Plant hormone signal transduction	K14484	XP_011649125.1	10.13	auxin-responsive protein IAA13, transcript variant X1
	K11816	XP_011649797.1	8.97	probable indole-3-pyruvate monooxygenase YUCCA4, transcript variant X1
	K14509	XP_031737453.1	−10.22	ethylene receptor 2, transcript variant X3

**Table 5 plants-14-03860-t005:** Transcriptional factors associated with unique DEGs in FG, TG, and SG plants.

**FG**			
**Protein ID**	**TF**	**Log_2_FC**	**Function**
XP_004133809.2	bHLH	1.56	transcription factor MTB1
XP_011648566.1	bHLH	10.54	transcription factor BIM1, transcript variant X1
XP_011657919.2	bHLH	9.29	transcription factor bHLH19
XP_011654291.1	bZIP	1.58	bZIP transcription factor 60
XP_011658569.1	bZIP	1.61	uncharacterized LOC101216189
NP_001267641.1	C2H2	1.35	zinc finger AN1 and C2H2 domain-containing stress-associated protein 11-like
XP_004141285.1	C2H2	3.76	zinc finger protein ZAT10
XP_004147699.1	C2H2	2.51	zinc finger protein ZAT12
XP_011658172.1	C2H2	6.97	uncharacterized LOC105435955
XP_011657270.2	CAMTA	10.93	calmodulin-binding transcription activator 5, transcript variant X1
XP_004134413.1	ERF	−4.69	ethylene-responsive transcription factor 14
XP_004136839.1	ERF	2.25	ethylene-responsive transcription factor 4
XP_004139438.1	ERF	1.47	ethylene-responsive transcription factor ERF039
XP_004140859.1	ERF	−1.10	ethylene-responsive transcription factor ERF118
XP_004144279.1	ERF	7.32	ethylene-responsive transcription factor ERF017
XP_011650877.1	G2-like	−11.43	protein PHR1-LIKE 2, transcript variant X2
XP_004149904.1	GATA	1.61	GATA transcription factor 8
XP_004145288.1	GRAS	2.19	scarecrow-like protein 34
XP_004140200.1	HB-other	9.64	uncharacterized LOC101207235, transcript variant X1
XP_004146371.1	HB-PHD	9.35	pathogenesis-related homeodomain protein, transcript variant X1
XP_004135200.1	HD-ZIP	−2.48	homeobox-leucine zipper protein HAT5
XP_031744925.1	MIKC_MADS	−2.86	MADS-box protein SVP, transcript variant X4
XP_004141899.1	MYB	2.26	transcription factor MYB44
XP_004147145.1	MYB	6.48	transcription factor MYB14
XP_031738637.1	MYB_related	8.11	hypothetical protein, transcript variant X2
XP_031739750.1	MYB_related	−7.80	telomere repeat-binding factor 1, transcript variant X4
XP_004139589.2	NAC	20.67	NAC domain-containing protein 40, transcript variant X1
XP_004149802.1	NAC	2.99	NAC domain-containing protein 2
NP_001292676.1	WRKY	7.92	probable WRKY transcription factor 26
XP_004134775.1	WRKY	2.42	probable WRKY transcription factor 31
XP_004149751.1	WRKY	6.03	probable WRKY transcription factor 41
XP_011652902.1	WRKY	1.92	probable WRKY transcription factor 7-like, transcript variant X1
**TG**			
**Protein ID**	**TF**	**Log_2_FC**	**Function**
XP_011651831.1	B3	10.11	B3 domain-containing transcription repressor VAL2, transcript variant X3
XP_004143425.1	bHLH	8.25	transcription factor bHLH18, transcript variant X1
XP_011660048.1	bHLH	7.82	transcription factor bHLH47, transcript variant X1
XP_031741810.1	C2H2	8.76	protein indeterminate-domain 5, chloroplastic-like, transcript variant X2
XP_011652639.1	HD-ZIP	8.41	homeobox-leucine zipper protein HDG5, transcript variant X1
XP_011657330.1	NAC	8.22	protein CUP-SHAPED COTYLEDON 3, transcript variant X1
XP_031741540.1	NAC	10.30	NAC domain containing protein 50, transcript variant X1
XP_011649248.1	TALE	−10.98	BEL1-like homeodomain protein 7, transcript variant X2
XP_031736434.1	YABBY	−3.35	axial regulator YABBY 5, transcript variant X2
**SG**			
**Protein ID**	**TF**	**Log_2_FC**	**Function**
XP_031743220.1	B3	−7.42	uncharacterized LOC101221625, transcript variant X32
XP_031736097.1	E2F/DP	8.47	E2F transcription factor-like E2FE, transcript variant X4
XP_011656418.1	ERF	−9.43	dehydration-responsive element-binding protein 2C, transcript variant X1
XP_031743007.1	NF-YA	−11.82	nuclear transcription factor Y subunit A-1, transcript variant X8
XP_011649247.1	TALE	−12.06	BEL1-like homeodomain protein 7, transcript variant X1
XP_031736725.1	TALE	−2.27	homeobox protein ATH1, transcript variant X7
XP_004136441.1	Trihelix	10.67	trihelix transcription factor ASR3, transcript variant X1
XP_004141520.1	Whirly	0.44	single-stranded DNA-binding protein WHY2, mitochondrial

## Data Availability

The data for this article can be found in the NCBI GenBank repository at https://www.ncbi.nlm.nih.gov/ under the BioProject PRJNA1320987. Raw sequence data for the BioSamples SAMN51192783-SAMN51192830 are deposited in SRA accessions SRR35280281-SRR35280235.

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
