# Peer review of "Molecular Pathways Associated with Cold Tolerance in Grafted Cucumber (Cucumis sativus L.)"

_plants, 2025, doi:10.3390/plants14243860_

Round 1
Reviewer 1 Report
Comments and Suggestions for Authors
Dear Authors!
In the peer-reviewed article, the authors studied the transcriptomic responses of cucumber heterografts and self-grafts under cold stress conditions in order to understand the molecular mechanisms that underlie the increased resistance provided by cold-tolerant rootstocks.
There are some comments for Authors.
- For some abbreviations, such as MAPK and ABA, the full names are not provided.
- In Chapter 2.1, which discusses experiments with plants, it would be helpful to include data on light intensity in addition to information about photoperiod and temperature.
- In the captions to Figures S1 and S2 in the Supplementary Materials, the temperature of the growing plants is given in degrees Celsius. However, the figures themselves use Fahrenheit as the unit of measurement for temperature.
- It would be helpful to include the Conclusions chapter in the article.
- In the References chapter, many references contain the names of organisms that should be written in italics.
After considering these comments, this manuscript can be recommended for publication in Plants.
Author Response
We sincerely thank the reviewers for their insightful comments. We have carefully addressed all suggestions, and the manuscript has been revised accordingly. These changes, highlighted in red, have strengthened the clarity and scientific quality of the manuscript.
- For some abbreviations, such as MAPK and ABA, the full names are not provided.
Full names are added for MAPK in line 290, and ABA in line 311.
- In Chapter 2.1, which discusses experiments with plants, it would be helpful to include data on light intensity in addition to information about photoperiod and temperature.
Thank you for the reviewer’s comment. We have added the light intensity 100 µmol m⁻² s⁻¹ in lines 118 in the revised version.
- In the captions to Figures S1 and S2 in the Supplementary Materials, the temperature of the growing plants is given in degrees Celsius. However, the figures themselves use Fahrenheit as the unit of measurement for temperature.
Thank you for the reviewer’s comment. Suggested changes have been made in supplementary figures.
- It would be helpful to include the Conclusions chapter in the article.
Thank you for the reviewer’s comment. Conclusion chapter has been added in lines 676-695 in the revised version.
- In the References chapter, many references contain the names of organisms that should be written in italics.
Thank you for the reviewer’s comment. The edits have been made in reference sections.
Reviewer 2 Report
Comments and Suggestions for Authors
This manuscript investigates the molecular mechanisms of using grafting to enhance cold tolerance in cucumber. The research topic is clear, and the experimental design is generally reasonable, combining biomass and transcriptome analyses. The research findings contribute to enriching the understanding of the molecular mechanisms of heterografting-induced cold tolerance in cucumber. However, the paper has some noticeable issues regarding the rigor of experimental design details, the depth of results and discussion, and the clarity of text writing and presentation, which weaken the reliability and persuasiveness of the conclusions.
1. The manuscript aims to elucidate how the 'rootstock' confers cold tolerance to the 'scion'. However, the experimental design includes 'ungrafted (UG)' plants as a control, which introduces a significant confounding variable. The grafting process itself is a form of mechanical damage or stress to the plants. When comparing heterografts (FG/TG) with ungrafted (UG) plants, the observed differences could stem from either the rootstock effect or the grafting wound stress, making the results impossible to interpret accurately. Based on the method description (2.5), all results in the manuscript are based on comparisons with UG, which is an extremely serious design flaw. A thorough restructuring of the entire paper is recommended, consistently comparing heterografts (FG/TG) with self-grafted (SG) plants. Both have undergone the grafting operation; the differences between them can be unambiguously attributed to the difference in rootstock genotype. Although the paper includes SG, this core comparison is not highlighted in many analyses and conclusions. Instead, frequent comparisons with UG weaken the rigor of the argument.
2. The paper remains at the level of describing 'which genes are up/down-regulated' and 'which pathways are enriched', but lacks sufficient biological explanation for how these changes specifically contribute to cold tolerance. For example, when discussing MAPK, TIFY, CPK and other genes, it only briefly mentions that they "may" be involved in cold tolerance, without proposing a specific and plausible regulatory model based on their known functions and their expression patterns in this experiment. The phenomenon of "early suppression, later recovery" of genes in hormone pathways also lacks in-depth physiological or ecological explanation. It is recommended that the discussion section delve deeper into interpreting the functional significance of the molecular changes and integrate the scattered molecular events into a plausible, hypothetical signaling network or regulatory model.
3. The first two paragraphs of the discussion section largely reiterate background information already mentioned in the introduction (such as the benefits of grafting), failing to directly delve into the analysis of this study's key findings, which reduces the academic value of the discussion.
4. The description of the healing environment is incomplete. While the photoperiod "18h light/6h dark" is mentioned, the light intensity is not specified, which is a key environmental parameter.
5. The text states "Each growth chamber had three blocks". It needs to be clearly stated whether each "block" is an independent replicate unit (e.g., each treatment had three independent pots/areas within each growth chamber) or a technical replicate. This is crucial for assessing the statistical significance of the experiment.
6. The sample size used for transcriptome analysis is severely insufficient, which is the most critical issue of the manuscript. The text states "Each biological replicate consisted of leaves from two plants" (Line 136). I understand this to mean that for each treatment, at each time point, within each growth chamber, one biological replicate is essentially a pooled sample from just two plants. More seriously, if these "two plants" came from the same "block", then the so-called "three biological replicates" may not be entirely independent. The aforementioned flaws seriously affect the reliability of all transcriptome data and subsequent analyses (e.g., DEG identification).
7. The rationale for sample selection is also insufficient. Discarding sampling for G1 solely based on the reason "high mortality rate" seems somewhat cursory. The dying tissues of G1 might provide unique molecular information about the limits of cold tolerance. If G1 plants almost all died, this should be stated explicitly, or G1 should be omitted entirely from the manuscript and only serve as the basis for the authors' preliminary experiments. It is also recommended that the authors supplement statistical data on mortality rates and conduct relevant comparative analyses to demonstrate that grafting indeed enhanced cold tolerance.
8. The interpretation of DEG analysis data lacks depth. For instance, the explanation for the counterintuitive phenomenon where the number of DEGs in the milder G3 environment is much higher than in the colder G2 environment is insufficient. This could be an important finding of this study but was overlooked.
9. Genes should be annotated with their species of origin and full name upon first mention.
10. The summary statement in lines 517-524 is overly broad. For example, "heterografts activate a complex regulatory network" is a conclusion, but there is almost no clear chain of evidence provided, making the conclusion seem vague.
In short, the manuscript suffers from a fundamental flaw in its core design. The transcriptomic analysis logic, which compares all grafted treatments solely against the ungrafted control, makes it impossible to discern whether the observed molecular responses stem from rootstock properties or the grafting wounding itself. This invalidates the supporting data for the paper's central conclusions. Furthermore, a critically inadequate sample size severely undermines the reliability of all research findings.
Comments on the Quality of English LanguageThe manuscript's English is suboptimal for academic publication, with numerous grammatical errors, awkward phrasing, and unprofessional word choices that hinder clarity. While the core scientific content is understandable, the language requires extensive editing to meet journal standards.
Author Response
We sincerely thank the reviewers for their insightful comments. We have carefully addressed all suggestions, and the manuscript has been revised accordingly. These changes, highlighted in red, have strengthened the clarity and scientific quality of the manuscript.
- The manuscript aims to elucidate how the 'rootstock' confers cold tolerance to the 'scion'. However, the experimental design includes 'ungrafted (UG)' plants as a control, which introduces a significant confounding variable. The grafting process itself is a form of mechanical damage or stress to the plants. When comparing heterografts (FG/TG) with ungrafted (UG) plants, the observed differences could stem from either the rootstock effect or the grafting wound stress, making the results impossible to interpret accurately. Based on the method description (2.5), all results in the manuscript are based on comparisons with UG, which is an extremely serious design flaw. A thorough restructuring of the entire paper is recommended, consistently comparing heterografts (FG/TG) with self-grafted (SG) plants. Both have undergone the grafting operation; the differences between them can be unambiguously attributed to the difference in rootstock genotype. Although the paper includes SG, this core comparison is not highlighted in many analyses and conclusions. Instead, frequent comparisons with UG weaken the rigor of the argument.
Thank you for raising this important point regarding the use of ungrafted (UG) plants as a control and the potential confounding effect of grafting-induced wound stress. We understand the reviewer’s concern and have clarified this more explicitly in the revised manuscript. Although UG plants were included to capture the full physiological response relative to a non-grafted baseline, all key interpretations related to rootstock-mediated effects were conducted by comparing heterografts (FG and TG) by parsing out the results from self-grafted (SG) plants, which control for grafting stress. To strengthen this further, we have added a dedicated section 3.6 presenting the unique DEGs identified specifically in heterografts, obtained after parsing out all DEGs shared with SG plants. This analytical step ensures that the DEGs highlighted as rootstock-responsive are not influenced by grafting wound effects but instead reflect true rootstock-specific transcriptional responses. Also, practical horticultural grafting inherently involves both grafting stress and rootstock-scion interactions, we believe that evaluating both SG-filtered unique DEGs and comparisons with UG provides a more comprehensive understanding of the combined and realistic effects experienced in grafted plants. We have revised the results and discussion to ensure that the SG-based contrasts are clearly emphasized and form the foundation of all mechanistic interpretations in lines in revised version.
- The paper remains at the level of describing 'which genes are up/down-regulated' and 'which pathways are enriched', but lacks sufficient biological explanation for how these changes specifically contribute to cold tolerance. For example, when discussing MAPK, TIFY, CPK and other genes, it only briefly mentions that they "may" be involved in cold tolerance, without proposing a specific and plausible regulatory model based on their known functions and their expression patterns in this experiment. The phenomenon of "early suppression, later recovery" of genes in hormone pathways also lacks in-depth physiological or ecological explanation. It is recommended that the discussion section delve deeper into interpreting the functional significance of the molecular changes and integrate the scattered molecular events into a plausible, hypothetical signaling network or regulatory model.
We appreciate the reviewer’s thoughtful comments regarding the need for deeper biological interpretation beyond listing DEGs and enriched pathways. In response, we have substantially expanded the interpretive component of the discussion and integrated a cohesive, hypothetical regulatory model that explains how the observed transcriptomic changes mechanistically contribute to cold tolerance in heterografted cucumber in conclusion section in revised version.
- The first two paragraphs of the discussion section largely reiterate background information already mentioned in the introduction (such as the benefits of grafting), failing to directly delve into the analysis of this study's key findings, which reduces the academic value of the discussion.
We appreciate the reviewer’s comment. However, we would like to clarify that the first two paragraphs of the discussion do not restate introductory background. Instead, they provide a concise overview of the major transcriptomic patterns observed in this study and contextualize these findings with previously reported grafting and cold-stress responses. This framing is important for setting up the subsequent, more detailed interpretation of specific pathways and molecular mechanisms. Because the discussion transitions directly from a high-level synthesis of our results to deeper pathway-level analyses, we believe the current structure supports a logical flow and enhances clarity. Therefore, we respectfully maintain the existing organization of the discussion while ensuring that the linkage between our findings and prior studies remains focused and relevant. Also, we have added discussion about the patterns of expression in different temperature zones in different rootstocks in lines 512-523 in revised version.
- The description of the healing environment is incomplete. While the photoperiod "18h light/6h dark" is mentioned, the light intensity is not specified, which is a key environmental parameter.
Thank you for the reviewer’s comment. We have added the light intensity 100 µmol m⁻² s⁻¹ in lines 118 in revised version.
- The text states "Each growth chamber had three blocks". It needs to be clearly stated whether each "block" is an independent replicate unit (e.g., each treatment had three independent pots/areas within each growth chamber) or a technical replicate. This is crucial for assessing the statistical significance of the experiment.
Thank you for pointing out the need for clearer definition of “blocks” within the growth chambers. We confirm that each block represents an independent biological replicate unit.
- The sample size used for transcriptome analysis is severely insufficient, which is the most critical issue of the manuscript. The text states "Each biological replicate consisted of leaves from two plants" (Line 136). I understand this to mean that for each treatment, at each time point, within each growth chamber, one biological replicate is essentially a pooled sample from just two plants. More seriously, if these "two plants" came from the same "block", then the so-called "three biological replicates" may not be entirely independent. The aforementioned flaws seriously affect the reliability of all transcriptome data and subsequent analyses (e.g., DEG identification).
Thank you for highlighting the concern regarding biological replication. We apologize for the lack of clarity in our original description and have revised the Methods section accordingly. In our experiment, one biological replicate consisted of pooled leaf tissue from two plants originating from the same block, and because each growth chamber contained three independent blocks, we obtained three biological replicates per treatment and time point, each derived from a different block. In our randomized complete block design, each block serves as an independent experimental unit, ensuring that the three biological replicates used for RNA-seq were truly independent despite pooling within blocks. Pooling two plants per replicate was necessary to obtain sufficient tissue for high-quality RNA extraction and to reduce within-block micro-environmental variation. Importantly, pooling 2–3 plants to generate a single biological replicate is a well-established practice in RNA-seq studies, especially when working with controlled-environment chambers where space and plant size limit the availability of tissue. We have clarified this sampling strategy in lines in revised version to prevent misinterpretation and to clearly communicate the independence and validity of our biological replicates.
- The rationale for sample selection is also insufficient. Discarding sampling for G1 solely based on the reason "high mortality rate" seems somewhat cursory. The dying tissues of G1 might provide unique molecular information about the limits of cold tolerance. If G1 plants almost all died, this should be stated explicitly, or G1 should be omitted entirely from the manuscript and only serve as the basis for the authors' preliminary experiments. It is also recommended that the authors supplement statistical data on mortality rates and conduct relevant comparative analyses to demonstrate that grafting indeed enhanced cold tolerance.
Suggested changes have been made in lines 132 in revised version
8. The interpretation of DEG analysis data lacks depth. For instance, the explanation for the counterintuitive phenomenon where the number of DEGs in the milder G3 environment is much higher than in the colder G2 environment is insufficient. This could be an important finding of this study but was overlooked.
We thank the reviewer for this comment. We have revised the discussion to provide a detailed interpretation of the DEG patterns across temperature regimes in lines 512-523. Specifically, heterografted plants exhibited a higher number of DEGs in the milder G3 conditions, reflecting active transcriptional reprogramming when stress remained within a physiologically manageable range. In contrast, self-grafted plants showed more DEGs in the colder G2 environment, largely driven by widespread downregulation rather than adaptive activation, indicating transcriptional suppression under more severe cold. These revisions explicitly address the rootstock-dependent differences in transcriptional plasticity and highlight the biological significance of the observed DEG patterns.
- Genes should be annotated with their species of origin and full name upon first mention.
We thank the reviewer for this helpful comment. While we agree that providing full gene names and species of origin is ideal, RNA-seq studies typically reference hundreds of genes in both the Results and Discussion sections. In such cases, full annotation for every gene is not commonly practiced, as it substantially disrupts readability and flow. Instead, we have ensured that all key genes central to the biological interpretation are fully annotated at first mention, while maintaining standard gene symbols for the remainder of the manuscript, consistent with other transcriptomics articles.
- The summary statement in lines 517-524 is overly broad. For example, "heterografts activate a complex regulatory network" is a conclusion, but there is almost no clear chain of evidence provided, making the conclusion seem vague.
Thank you for the reviewer’s insightful comment. The newly added content in the revised section now provides the necessary mechanistic interpretation to support the statements made in the discussion. Furthermore, the expanded interpretation also strengthens the conclusion section, which now reflects a cohesive summary of the proposed signaling network and more clearly articulates how heterografting mediates enhanced cold resilience.
Round 2
Reviewer 2 Report
Comments and Suggestions for Authors
The authors have provided thoughtful and comprehensive responses to most of the comments, and the manuscript has undergone substantive revisions. The depth of biological interpretation in the Discussion and the overall coherence of the Conclusion have been notably strengthened. However, there remains room for improvement in the standardization of literature citations and the clarity of language expression. For example, references are not cited with sufficient clarity. The citation [14–21] in line 44, where so many references are listed for a single sentence, demonstrates poor specificity.
Author Response
The authors have provided thoughtful and comprehensive responses to most of the comments, and the manuscript has undergone substantive revisions. The depth of biological interpretation in the Discussion and the overall coherence of the Conclusion have been notably strengthened. However, there remains room for improvement in the standardization of literature citations and the clarity of language expression. For example, references are not cited with sufficient clarity. The citation [14–21] in line 44, where so many references are listed for a single sentence, demonstrates poor specificity.
We thank the reviewer for acknowledging the substantial improvements made to the Discussion and Conclusion sections. We also appreciate the constructive feedback regarding citation clarity and language expression.
In response to the concern about the non-specific citation range [14–21] in line 44, we have carefully revised the manuscript to ensure that each statement is supported by the most relevant and specific references. The original broad citation range has been replaced with individual, directly applicable citations. We also conducted a thorough review of the entire manuscript to improve the precision, consistency, and formatting of all references.
Additionally, we have further refined the language throughout the manuscript to enhance clarity and readability.
In response to the reviewer’s comments on figure quality, we have improved all figures for clarity and resolution and provided separate high-quality figure files as required by the journal.